# Effects of Intrauterine Growth Restriction (IUGR) on Growth and Body Composition Compared to Constitutionally Small Infants

**DOI:** 10.3390/nu15194158

**Published:** 2023-09-26

**Authors:** Elisabeth Calek, Julia Binder, Pilar Palmrich, Felix Eibensteiner, Alexandra Thajer, Theresa Kainz, Karin Harreiter, Angelika Berger, Christoph Binder

**Affiliations:** 1Division of Neonatology, Pediatric Intensive Care Medicine and Neuropediatrics, Department of Pediatrics and Adolescent Medicine, Medical University of Vienna, 1090 Vienna, Austria; elisabeth.calek@meduniwien.ac.at (E.C.); alexandra.thajer@meduniwien.ac.at (A.T.); theresa.kainz@meduniwien.ac.at (T.K.); karin.harreiter@meduniwien.ac.at (K.H.); angelika.berger@meduniwien.ac.at (A.B.); 2Department of Obstetrics and Gynecology, Medical University of Vienna, 1090 Vienna, Austria; julia.binder@meduniwien.ac.at (J.B.); pilar.palmrich@meduniwien.ac.at (P.P.); 3Department of Emergency Medicine, Medical University of Vienna, 1090 Vienna, Austria; felix.eibensteiner@meduniwien.ac.at

**Keywords:** air displacement plethysmography, body composition, nutritional management, preterm, intrauterine growth restriction, small for gestational age, appropriate for gestational age

## Abstract

(1) Intrauterine growth restriction (IUGR) is associated with multiple morbidities including growth restriction and impaired neurodevelopment. Small for gestational age (SGA) is defined as a birth weight <10th percentile, regardless of the etiology. The term is commonly used as a proxy for IUGR, but it may represent a healthy constitutionally small infant. Differentiating between IUGR and constitutionally small infants is essential for the nutritional management. (2) Infants born at <37 weeks of gestation between 2017 and 2022, who underwent body composition measurement (FFM: fat-free mass; FM: fat mass) at term-equivalent age, were included in this study. Infants with IUGR and constitutionally small infants (SGA) were compared to infants appropriate for gestational age (AGA). (3) A total of 300 infants (AGA: *n* = 249; IUGR: *n* = 40; SGA: *n* = 11) were analyzed. FFM (*p* < 0.001) and weight growth velocity (*p* = 0.022) were significantly lower in IUGR compared to AGA infants, but equal in SGA and AGA infants. FM was not significantly different between all groups. (4) The FFM Z-score was significantly lower in IUGR compared to AGA infants (*p* = 0.017). Being born constitutionally small compared to AGA had no impact on growth and body composition. These data showed that early aggressive nutritional management is essential in IUGR infants to avoid impaired growth and loss of FFM.

## 1. Introduction

Intrauterine growth restriction (IUGR) is well known to be associated with increased neonatal morbidity and mortality [1,2,3]. A recent systematic review and meta-analysis showed that IUGR is linked to neurodevelopmental delays up to 12 years of age [4]. As achieving adequate growth for infants with IUGR remains a major challenge for neonatologists worldwide, early aggressive nutritional management after birth is essential [5]. Extrauterine growth restriction (EUGR) is common among preterm infants with IUGR. It is associated with nutritional and metabolic deficits, especially during the first weeks of life, and with neurodevelopmental delays [3,6]. Infants with IUGR often require prolonged intravenous and enteral supplementation due to limited feeding tolerance, inadequate nutrient reserves, and tissue storage [7]. Early adequate nutritional supply in infants with IUGR is crucial for normal growth and brain development [6,8]. Therefore, the major nutritional goal in these infants is to achieve adequate nutritional intake to avoid postnatal growth restriction, which is associated with impaired neurodevelopment [9,10,11]. Growth, usually measured by anthropometric parameters including weight, length, and head circumference, has been shown to be a good marker to evaluate nutritional management and status [12]. Furthermore, body composition measurement is a valuable tool to assess quantitative growth by measuring fat-free mass (FFM) and fat mass (FM) [13]. A recent study has demonstrated that FFM is a better measure of optimal brain growth compared to anthropometric parameters at term-equivalent age [14]. Deficits in FFM at term-equivalent age have also been shown to be associated with impaired neurodevelopment [15]. Consequently, adequate body composition, especially FFM, is essential for normal brain maturation and development [16].

IUGR is defined as a fetus that has failed to reach its growth potential, which can be due to several conditions, but is primarily caused by placental insufficiency and is characterized by impaired oxygen and nutritional supply [17,18]. The underlying cellular and molecular mechanism of fetal growth restriction is still not fully investigated [17,19]. Placental-mediated fetal growth restriction arises mainly from maldevelopment of the placental vascular system [20]. Several studies showed that impaired fetoplacental angiogenesis is associated with fetal growth restriction [21,22]. Furthermore, recent research investigated novel integrin–extracellular matrix interactions that regulate placental angiogenesis in severe fetal growth restriction [19]. However, further research is needed to investigate the underlying cause of fetal growth restriction as well as prevention and treatment strategies. The diagnosis is based on prenatal fetal biometry and Doppler ultrasound parameters [17]. On the contrary, small for gestational age (SGA) is defined as a birth weight below the 10th percentile for gestational age, regardless of the etiology [23,24]. The term is often used in the literature as a proxy for IUGR [25,26,27], from which it should be clearly distinguished, as IUGR refers to a pathological condition and does not represent a constitutionally small infant [28]. Constitutionally small infants, defined as SGA at birth, present with a birth weight less than the 10th percentile due to the following factors: parental weight, height, and ethnicity [29,30]. These infants may not be at increased risk for perinatal mortality and morbidity, including EUGR [31].

The difference between infants with IUGR and constitutionally small infants has not been considered in neonatal treatment concepts [32,33]. IUGR infants may need an early aggressive nutritional supply to achieve appropriate growth, but it is not clear whether this also applies to constitutionally small infants [5,34,35]. The European Society for Paediatric Gastroenterology, Hepatology and Nutrition (ESPGHAN) recommends feeding enhanced nutrients up to the 52nd week of gestation in all infants below the 10th percentile at term-equivalent age, regardless of whether they are defined as IUGR or constitutionally small [36].

To the best of our knowledge, there are no studies evaluating growth and body composition in preterm infants with IUGR in comparison to constitutionally small infants, which might have a relevant impact on future nutritional care and strategies of these infants. The aim of this study was to evaluate growth and body composition at term-equivalent age with a consideration of nutritional management in IUGR and constitutionally small infants compared to preterm infants appropriate for gestational age (AGA).

## 2. Materials and Methods

### 2.1. Study Design

This exploratory retrospective cohort study was performed at the Department of Pediatrics and Adolescent Medicine, Division for Neonatology, Intensive Care Medicine and Neuropediatrics at the Medical University of Vienna, Austria. The study was approved by the Ethics Committee of the Medical University of Vienna (Number: 1602/2019). The primary objective of this study was to evaluate growth parameters including weight, length, and head circumference at birth and at term-equivalent age as well as a body composition measurement (FFM and FM) at term-equivalent age in constitutionally small infants (SGA group) and preterm infants with IUGR (IUGR group) in comparison to a control group of preterm infants appropriate for gestational age (AGA group). The study was conducted at a level IV neonatal intensive care unit.

### 2.2. Patient Groups

Preterm infants born before 37 weeks of gestation between the years 2017 and 2022, who underwent air displacement plethysmography (PEA POD^®^, Cosmed, Concord, CA, USA) to determine body composition (FFM and FM) at term-equivalent age, were enrolled in the study. Exclusion criteria were chromosomal abnormalities or genetic and metabolic disorders. In our hospital, body composition measurements are performed in all preterm infants at term-equivalent age as part of our standard clinical practice. Group assignment (AGA, IUGR, and SGA groups) was based on weight percentile and signs of placental insufficiency determined by prenatal ultrasound [17,37]. In singleton pregnancies, IUGR was diagnosed according to the consensus definition by Gordjin et al. [17], which is defined as follows: (1) estimated fetal weight (EFW) <10th percentile and (2) uterine artery pulsatility index (Uta-PI) and/or umbilical artery PI (UA-PI) >95th percentile and/or middle cerebral artery (MCA-PI) <5th percentile. Constitutionally small (SGA group) was defined as infants with a birth weight <10th percentile and without any signs of placental insufficiency determined by prenatal ultrasound. AGA was defined as birth weight between the 10th and the 90th percentile without Doppler ultrasound abnormalities [17]. In multiple pregnancies, IUGR was defined according to the consensus definition by Khalil et al. [37]. For monochorionic twin pregnancies, at least two out of four parameters were required for diagnosis: EFW and/or abdominal circumference (AC) of one twin <10th percentile, EFW discordance of ≥25%, and UA-PI of the smaller twin >95th percentile. In dichorionic twin pregnancies, at least two out of three parameters were required for diagnosis: EFW of one twin <10th percentile, EFW discordance of ≥25%, and UA-PI of the smaller twin >95th percentile. Intertwin EFW discordance was calculated as 100 × (A −  B)/A, where A is the EFW of the larger fetus and B the EFW of the smaller one [37]. Infants with feto-fetal transfusion syndrome were excluded from this analysis, due to a risk of bias [38].

### 2.3. Growth and Body Composition

Air-displacement plethysmography (ADP) belongs to the two-compartment model of body composition that measures the density and volume of the subject and provides information about FFM and FM [13]. Using the so-called PEA POD^®^ (COSMED, Concord, CA, USA), body composition can be carried out in less than five minutes and without anesthesia [39]. The advantages of the ADP are accurate and reliable results using a very simple and non-invasive method [40]. Preterm infants underwent ADP to assess body composition at term-equivalent age, as part of our standard clinical practice. Based on published reference charts, sex- and age-adjusted Z-Scores were calculated for FFM and FM [39]. FFM and FM percentages, FFM and FM kilograms, and FFM and FM Z-Scores are presented.

Data on growth including weight, length, and head circumference were obtained retrospectively and analyzed from birth until term-equivalent age. Weight (g/kg/day), length (cm/kg/day), and head circumference (cm/kg/day) gain velocity were calculated from birth until term-equivalent age. Z-scores and percentiles for weight, length, and head circumference were calculated using Fenton growth charts [41] and WHO growth charts [42].

Body weight was measured daily with an electronic scale and when the infant reached 1000 g, body weight was measured every 48 h. In addition, supine length was measured using an infant length board, and head circumference was determined using a flexible cloth measuring tape.

### 2.4. Neonatal Morbidities

Intraventricular hemorrhage (IVH) was defined according to Papile et al. [43] and necrotizing enterocolitis (NEC) according to Bell et al. [44]. Retinopathy of prematurity (ROP) was defined according to The International Committee for the Classification of ROP [45] and bronchopulmonary dysplasia (BPD) was defined as an oxygen requirement of >21% at a gestational age of 36 + 0 weeks [46]. Culture-proven sepsis was defined as a positive result for one or more bacterial or fungal cultures obtained from the blood of the infants with clinical signs of infection or treated with appropriate antibiotics for five or more days. The duration of mechanical ventilation in days was evaluated. Demographic and perinatal data included gestational age at birth, birth weight, sex, antenatal steroid therapy, mode of delivery, preterm premature rupture of the membrane (PPROM), APGAR Scores after 5 and 10 min, umbilical artery pH, and surfactant application after birth.

### 2.5. Nutritional Management

Enteral nutrition with mother’s milk or, if not available, pasteurized preterm single donor milk (holder pasteurization) or preterm formula (in infants >32 weeks of gestation) was started on the first day of life and increased by 20–30 mL/kg/day. Fortification of the enteral nutrition with bovine-based fortifiers (Aptamil FMS—Milupa Nutricia GmbH, Frankfurt, Germany) in infants >26 weeks of gestation or human milk-based fortifier (Humavant +6—Prolacta Bioscience, Duarte, CA, USA) in infants <26 weeks of gestation occurred at an enteral intake of 100 mL/kg/day. Preterm infants also received parenteral nutrition starting on the first day of life. Carbohydrate, protein, and fat intake was administered according to the recommendations of ESPGHAN [47]. Parenteral nutritional supply was discontinued when enteral nutritional intake reached 140–160 mL/kg/day. The on parenteral nutrition in days and the type of nutrition are reported. The enteral nutrition (mother’s own milk, fortification, formula, and mixed nutrition) at discharge and parenteral nutritional supply (kilocalories kcal/kg/d, carbohydrates mg/kg/min, proteins g/kg/d, and fat g/kg/d) are also reported.

### 2.6. Statistics

The demographic information and descriptive statistics were presented using the frequency distribution, median, and interquartile range (IQR). The weight, length, head circumference, FFM, and FM measurements were standardized into Z-scores using the LMS method, based on growth charts specific to sex and gestational age [36]. A multivariable regression analysis was used to investigate the association between body composition (FFM gram, FM gram, and FFM and FM Z-Scores) and body weight at term-equivalent age and the study groups (AGA, IUGR, and SGA) while adjusting for the covariates, including sex [48], gestational age at birth [49], and age, weight, and length at measurement to adjust for body size [49]. Mann–Whitney-U and Pearson’s chi-square tests were used to compare baseline characteristics, growth velocity (grams/kg/day from birth until discharge), nutritional management (parenteral and enteral nutrition), and co-morbidities (IVH, NEC, ROP, BPD, and culture-proven sepsis). The data were analyzed using the software Statistical Package for Social Science (SPSS) version 28 for Mac (IBM Corp, Armonk, New York, NY, USA). A *p*-value of less than 0.05 was considered statistically significant.

## 3. Results

The analysis included a total of 300 premature infants (AGA: *n* = 249, IUGR: *n* = 40, and SGA: *n* = 11). The body composition measurement was not performed in four infants due to following conditions: continuous oxygen requirement: *n* = 1, IUGR group; hemodynamic instability: *n* = 1, AGA group; and loss of follow up: *n* = 2, AGA group (Figure 1). Table 1 displays the baseline characteristics. The median gestational age in weeks at birth was significantly higher in IUGR infants (30.4; IQR: 25.9, 35.7) in comparison to AGA infants (26.9; IQR: 25.8, 31.0), (*p* < 0.001), and not significantly different between SGA infants (28.1; IQR: 25.5, 28.9) and AGA infants (26.9; IQR: 25.8, 31.0), (*p* = 0.21). The anthropometric parameters (weight, length, and head circumference Z-Scores) at birth were significantly lower in SGA compared to AGA infants (*p* ≤ 0.001, respectively) and significantly lower in IUGR compared to AGA infants (*p* ≤ 0.001, respectively), as expected (Table 1). The mode of delivery, APGAR Scores at 5 and 10 min, and umbilical artery pH values were not significantly different between the three study groups.

The anthropometric parameters at discharge, growth velocities from birth until discharge, and short-term outcome parameters are shown in Table 2. The median gestational age in weeks at body composition measurement was not significantly different between the groups: AGA 42.1 (IQR: 40.0, 46.3), IUGR 43.0 (IQR: 40.5, 46.9), and SGA 42.1 (IQR: 39.4, 52.6); IUGR versus AGA (*p* = 0.29) and SGA versus AGA (*p* = 0.09). At discharge, the median weight (*p* < 0.001) and length (*p* < 0.001) were significantly lower in infants in the IUGR group compared to the AGA group. The head circumference was not significantly different between the IUGR and AGA groups (*p* = 0.10). There were no significant differences in anthropometric parameters including weight, length, and head circumference at discharge between the SGA and AGA groups (Table 2). Short-term outcome parameters including ROP, IVH, NEC, BPD, and culture-proven septicemia were not significantly different between the three study groups (Table 2). The weight growth velocity (g/kg/day) was significantly lower in infants in the IUGR group compared to infants in the AGA group (*p* = 0.022). The weight growth velocity was not significantly different between the SGA and AGA groups (*p* = 0.74). There was no statistically significant difference in the median increase of length and head circumference from birth until discharge between the IUGR and AGA groups (*p* = 0.14 and *p* = 0.35, respectively) and the SGA and AGA groups (*p* = 0.97 and *p* = 0.22, respectively) (Table 2).

The nutritional parameters are displayed in Table 3. The median number of days on parenteral nutrition was not significantly different between the groups (IUGR versus AGA, *p* = 0.68, and SGA versus AGA, *p* = 0.07). The enteral and parenteral nutritional supply (kcal, carbohydrates, protein, and fat) were not significantly different between the groups (Table 3). The enteral nutrition at discharge was not significantly different between the groups (Table 3). There was no significant difference between the three groups with regard to exclusive breast milk feeding at discharge (IUGR versus AGA, *p* = 0.68, and SGA versus AGA, *p* = 0.45).

Table 4 displays the unadjusted anthropometric parameters (weight, length, and head circumference) as well as the body composition measurements (FFM and FM) at term-equivalent age. The age at body composition measurement was not significantly different between the groups: IUGR versus AGA, *p* = 0.38, and SGA versus AGA, *p* = 0.91.

A linear regression model was used to analyze the association between body composition (FFM and FM Z-Scores) and weight at term-equivalent age and the study groups (AGA, IUGR, and SGA), while adjusting for the covariates such as sex, gestational age at birth, and age, weight, and length at measurement to adjust for body size. The FFM Z-score and FFM gram were significantly lower in infants with IUGR in comparison to AGA infants (*p* = 0.017 and *p* < 0.001, respectively), while there were no significant differences in the FFM Z-Score and FFM gram between SGA and AGA infants (*p* = 0.78 and *p* = 0.09, respectively) (Table 5). The FM Z-Scores and FM gram were not significantly different between the IUGR and AGA groups or between the SGA and AGA groups (Table 5). Sex did not significantly influence weight and body composition. Weight: IUGR versus AGA, *p* = 0.12, and SGA versus AGA, *p* = 0.23. FFM gram: IUGR versus AGA, *p* = 0.12, and SGA versus AGA, *p* = 0.10. FFM Z-Score: IUGR versus AGA, *p* = 0.06, and SGA versus AGA, *p* = 0.07. FM gram: IUGR versus AGA, *p* = 0.24, and SGA versus AGA, *p* = 0.21. FM Z-Score: IUGR versus AGA, *p* = 0.13, and SGA versus AGA, *p* = 0.14.

## 4. Discussion

This study evaluated growth and body composition at term-equivalent age in preterm infants with IUGR and constitutionally small infants in comparison to AGA infants. The FFM Z-Score and FFM gram were significantly lower in infants with IUGR in comparison to AGA infants, while there were no significant differences in FFM Z-Scores and FFM gram between SGA and AGA infants. The FM Z-Scores and FM gram were not significantly different in IUGR and SGA infants compared to AGA infants. The weight growth velocity from birth to discharge was significantly lower in the IUGR group than in the AGA group. There were no significant differences in growth velocity between the SGA and AGA groups.

These data demonstrate that the weight growth and FFM were significantly lower in IUGR infants compared to AGA infants at term-equivalent age. Being constitutionally small compared to AGA had no significant impact on growth and body composition at term-equivalent age. The study underlines the importance of distinguishing between IUGR and infants being born constitutionally small. Moreover, these results are essential for the nutritional treatment concepts in constitutionally small infants, and aggressive nutritional management can be refrained from in these infants.

The terms IUGR and SGA are often used synonymously in newborns, but the underlying pathomechanism and etiology of these patterns are related to different growth [50,51]. IUGR infants are newborns who fail to reach their potential growth mainly due to a placental insufficiency, resulting in impaired placental oxygen and nutritional transport [50,52]. Placental insufficiency has an impact on glucose sensitivity and placental transport capacity, resulting in fetal hypoglycemia [53,54]. Importantly, they lead to impaired fetal growth and fetal metabolism [55]. Furthermore, IUGR is associated with protein synthesis inhibition and impaired skeletal muscle growth [56,57]. It is well known that the accretion of lean mass is dependent on protein synthesis [57]. Previous studies demonstrated that lean mass is decreased in infants with IUGR compared to AGA infants and is associated with impaired growth [55]. EUGR is still a major concern in infants with IUGR and previous studies have shown that impaired growth and loss in lean mass is associated with impaired neurodevelopment [58]. An aggressive early nutritional supply and management are essential to avoid EUGR [5]. Furthermore, the nutritional goal is to achieve an early catch-up growth to decrease the risk of impaired neurodevelopment [10,15]. To improve nutritional outcomes, it is crucial to focus on monitoring and analyzing body composition rather than relying solely on weight measurements [13]. This highlights the need for a special focus on future nutritional management strategies for infants with IUGR, perhaps through the implementation of individually fortified diets based on body composition results. Previous studies demonstrated that preterm infants do not receive sufficient nutritional intake under the currently recommended nutritional management, especially standard fortification [59,60]. In a randomized controlled trial, Rochow et al. [61] found that individualized target fortification by adjusting breast milk nutrients compared to standard fortification was associated with a higher macronutrient intake, weight gain, and body weight at discharge in the individualized target fortification group. Furthermore, Parat et al. [62] showed that the use of targeted fortification strategies was associated with a higher protein intake and FFM at term-equivalent age, which might be important to avoid growth failure in IUGR infants. Individualized target fortification might be an additional important nutritional strategy in all preterm infants, but especially in IUGR infants, it may prevent growth failure and co-morbidities. The fortification of human milk should be considered in IUGR infants to achieve adequate nutrient supply, growth, and neurodevelopment.

Optimized nutritional management, quality of growth, and neurological outcomes are highly related to each other. Bruno et al. [25] showed that IUGR infants have significantly smaller thalamus and basal ganglion volumes compared to AGA infants. Furthermore, Binder et al. [14] demonstrated that the FFM Z-Score serves as a more reliable indicator for optimal brain growth compared to anthropometric parameters. Therefore, body composition at term-equivalent age is a good prognostic parameter for neonatal neurodevelopment and the primary nutritional goal is to achieve an optimal nutritional intake to prevent postnatal growth restriction, particularly the loss of FFM [63,64].

The ESPGHAN recommends feeding enhanced nutrients up to the 52nd week of gestation in all infants below the 10th percentile at term-equivalent age, regardless of the pathology [65]. However, constitutionally small infants may not require an additional nutrient supply directly after birth and at term-equivalent age [35]. These infants may have appropriate growth after birth [50]. Constitutionally small infants may also not be at higher risk of neonatal growth restriction [50]. Therefore, it is still unclear and has not been investigated so far, if constitutionally small infants may need aggressive nutrition for optimum growth and neurodevelopment. Previous studies have shown that aggressive nutritional management and accelerated growth may lead to a high risk of obesity and cardiovascular diseases in later life [66,67]. Furthermore, the additional nutrient supply may lead to an early catch-up growth and higher FM, measured by body composition, which is associated with obesity and metabolic syndrome [65,68,69].

In our study, the weight at birth was significantly lower in SGA infants in comparison to AGA infants. Importantly, the weight growth velocity from birth until term-equivalent age and body composition, especially FFM, were significantly lower in IUGR compared to AGA infants. The median FFM Z-Score was −0.6 lower in the IUGR group in comparison to the AGA group at term-equivalent age. These data are consistent with the literature and underline that adequate nutrition is essential to avoid EUGR [10,15]. Poor weight gain and loss in FFM are well known to be associated with impaired neurodevelopment [15,58]. Follow-up assessments after discharge are important in these infants to prevent long-term consequences. The adequate nutritional management in IUGR infants has not been fully investigated, but this study demonstrated that the current recommendations [47] are not sufficient to prevent growth failure. The nutritional management was according to the current ESPGHAN recommendation [47]. The nutritional management, nutrient supply, and days on parenteral nutrition during hospital stay were not significantly different between the three study groups. Furthermore, the rates of exclusively mother’s own milk at discharge were not significantly different between the groups (between 70 and 80%). These data support the hypothesis that the nutritional supply was inadequate to support normal growth in infants with IUGR. Consequentially, individualized fortification may improve growth and body composition in infants with IUGR. However, further research is needed to investigate if the nutritional supply during pregnancy and consequently the growth in utero may be the dominant factors in limiting postnatal growth and body composition.

On the contrary, the growth velocity and body composition, especially FFM, are not significantly different between constitutionally small infants and AGA. Several studies explored the optimum nutritional management, especially protein intake, to avoid growth restriction [9,10,11]. We showed that constitutionally small infants received adequate nutritional intake following ESPGHAN guidelines to gain similar growth and body composition to AGA infants. These results are significant and emphasize that constitutionally small infants have normal and adequate growth. Studies on growth and body composition especially on long-term development are needed. Ongoing follow-up studies in these infants will show the long-term effects on growth, body composition, and neurodevelopment.

A recent systematic review and meta-analysis evaluated growth and body composition in primarily term infants with IUGR and infants born SGA compared to AGA infants [58]. Three studies [70,71,72] with a relatively small sample size of preterm infants and late preterm infants were included in the analysis. One study [70] compared body composition measured by dual-energy X-ray absorptiometry (DEXA) at term-equivalent age in IUGR infants (*n* = 14) with that of AGA infants (*n* = 68). FFM was significantly lower in IUGR infants and these results are consistent with our study. These data emphasize that EUGR is still a significant challenge for neonatologists, and early aggressive adequate nutritional management is important to avoid long-term morbidities in these infants. However, the optimum nutritional management in IUGR infants is still unclear. Randomized controlled trials with nutritional interventions are needed to investigate adequate nutritional intake in these infants. However, two studies [71,72] in the systematic review evaluated body composition in SGA infants (*n* = 46) compared to AGA infants (*n* = 165). Notably, SGA was defined as birth weight below the 10th percentile, and no distinction between IUGR and constitutionally small infants at birth was performed. As already discussed in detail, the distinction between IUGR and constitutionally small infants is of major importance and should be diligently carried out. Therefore, a comparison between the meta-analysis and our study is not appropriate. In our study, a differentiation between IUGR infants and constitutionally small infants was well performed and further studies are necessary to evaluate the long-term effect on growth, body composition, and neurodevelopment. Interestingly, short-term outcomes including ROP, NEC, IVH, culture-proven sepsis, and BDP were not significantly different between the groups, especially not between IUGR and AGA infants. Previous studies showed that IUGR infants have a higher risk of neonatal morbidities [1,2,3]. However, co-morbidities including ROP, NEC, IVH, culture-proven sepsis, and BDP are inversely related to gestational age at birth. In our study, the median age at birth was significantly higher in IUGR infants (30.4 weeks) in comparison to AGA infants (26.9 weeks) (*p* < 0.001). We postulated that this observation might explain these results.

One limitation of the study is the relatively small sample size and the retrospective nature of this study. We considered a matching process, but the sample size in this study is relatively small, because being constitutionally small is an exceedingly rare condition and gestational age at birth varied widely. Therefore, we incorporated gestational age at birth as a covariate in the multivariable regression analysis. However, designing and conducting a clinical trial with an adequate number of constitutionally small infants would be very challenging. Furthermore, these data are generated from an exploratory study and the analysis requires cautious interpretation. The sample size of infants in the SGA group (*n* = 11) is relatively low and the comparison with the AGA group should be interpreted with caution. Further studies with a larger sample size and power calculation are necessary to draw final conclusions. However, this is the first study that distinguished between IUGR infants and constitutionally small infants, and evaluated the effect on growth and body composition. Furthermore, this study provides novel insights into growth and body composition in constitutionally small infants. Follow-up neurodevelopmental assessments using the Bayley Scales of Infants and Toddler Development at two and three years of age, as well as the Kaufman Assessment Battery of Children at five and a half years of age, will provide insight into growth and long-term neurodevelopment. A strength of our study is the use of body composition to accurately measure qualitative growth (FFM and FM) at term-equivalent age.

## 5. Conclusions

This study found that the weight growth velocity from birth to discharge was significantly lower in the IUGR group than in the AGA group. The FFM Z-Score and FFM gram were significantly lower in infants with IUGR in comparison to AGA infants, while there were no significant differences in FFM Z-Scores and FFM gram between SGA- and AGA infants. The weight growth velocity from birth until discharge was not significantly different between healthy constitutionally small infants and AGA infants. We confirm that growth velocity and FFM are suitable parameters on which to base the nutritional management of neonates. Our data show that the differentiation of IUGR and constitutionally small infants is important for nutritional management, especially for infants with IUGR. Early aggressive nutritional management is essential in IUGR infants to achieve optimum growth and development but might not be beneficial for constitutionally small infants.

## Figures and Tables

**Figure 1 nutrients-15-04158-f001:**
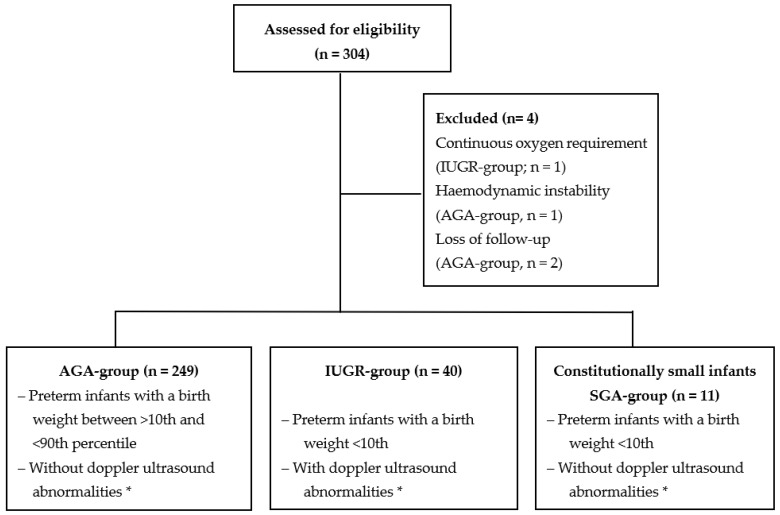
Overview of the study groups. * Gordijn, S.J. et al., 2016, [17].

**Table 1 nutrients-15-04158-t001:** Baseline characteristics.

Variables	AGA Group (*n* = 249)	IUGR Group (*n* = 40)	SGA Group (*n* = 11)	AGA vs. IUGR *p*-Values	AGA vs. SGA *p*-Values
Gestational age, weeks *	26.9 (25.8, 31.0)	30.4 (25.9, 35.7)	28.1 (25.5, 28.9)	<0.001	0.21
Male, % (*n*)	58 (144/249)	35 (14/40)	64 (7/11)	0.006	0.56
Antenatal steroids, % (*n*)	43 (107/249)	58 (23/40)	82 (9/11)	0.13	0.016
PROM, % (*n*)	17 (41/249)	5 (2/40)	18 (2/11)	<0.001	0.21
Caesarean delivery, % (*n*)	69 (171/249)	83 (33/40)	73 (8/11)	0.08	0.88
APGAR Score, 5 min *	9 (8, 9)	9 (9, 9)	9 (9, 9)	0.11	0.26
APGAR Score, 10 min *	9 (9, 9)	9 (9, 10)	9 (9, 9)	0.83	0.83
Umbilical artery, pH *	7.32 (7.29, 7.39)	7.30 (7.27, 7.36)	7.31 (7.28, 7.34)	0.07	0.51
Anthropometry at birth					
Birth weight, gram *	935 (757, 1399)	1050 (550, 1835)	623 (543, 735)	0.05	<0.001
Birth weight, Z-Score *	0.0 (−0.8, 0.3)	−1.6 (−1.8, −1.5)	−1.5 (−2.0, −1.4)	<0.001	<0.001
Birth length, cm *	35 (32, 41)	39 (30, 44)	31 (30, 33)	0.09	<0.001
Birth length, Z-Score *	0.0 (−0.7, 0.9)	−1.2 (−1.9, −0.7)	−1.7 (−2.5, −1.4)	<0.001	<0.001
Birth HC, cm *	24.8 (23.0, 29.0)	25.5 (21.3, 30.5)	23.3 (21.5, 25.4)	0.89	0.001
Birth HC, Z-Score *	0.2 (−0.3, 0.9)	−1.3 (−1.9, −0.9)	−1.1 (−2.0, −0.1)	<0.001	<0.001

* Values are median (interquartile range). Head circumference (HC), preterm premature rupture of the membrane (PPROM).

**Table 2 nutrients-15-04158-t002:** Growth velocity (from birth until discharge), short-term outcome parameters, and nutritional management.

Variables	AGA Group (*n* = 249)	IUGR Group (*n* = 40)	SGA Group (*n* = 11)	IUGR vs. AGA *p*-Values	SGA vs. AGA *p*-Values
PMA at discharge, week *	38.1 (36.9, 40.0)	39.7 (37.7, 41.5)	39.7 (37.6, 45.4)	0.29	0.09
Discharge weight, gram *	2805 (2442, 3114)	2333 (1998, 2765)	2730 (2080, 3190)	<0.001	0.67
Weight velocity, g/kg/d *	22.0 (19.5, 24.5)	20.3 (18.5, 23.1)	21.5 (17.7, 24.1)	0.022	0.74
Discharge length, cm *	46 (44, 48.5)	44 (41, 46)	44 (40, 46)	<0.001	0.07
Length increase, cm/week *	0.95 (0.82, 1.07)	1.03 (0.91, 1.10)	0.94 (0.83, 1.11)	0.14	0.97
Discharge HC, cm *	33.0 (31.5, 34)	32.0 (30.8, 33.8)	31.5 (30.7, 33.8)	0.10	0.37
HC increase, cm/week *	0.69 (0.59, 0.78)	0.72 (0.61, 0.81)	0.74 (0.58, 0.83)	0.35	0.22
ROP grade II-IV, % (*n*)	13 (33/249)	10 (4/40)	27 (3/11)	0.43	0.17
IVH (stage ≥ 2), % (*n*)	7 (18/249)	8 (3/40)	18 (2/11)	0.90	0.18
NEC (stage ≥ 2), % (*n*)	8 (19/249)	3 (1/40)	9 (1/11)	0.23	0.78
BPD, % (*n*)	12 (30/249)	10 (4/40)	27 (3/11)	0.63	0.13
Culture-proven sepsis, % (*n*)	22 (55/249)	33 (13/40)	27 (3/11)	0.16	0.78

* Values are median (interquartile range). Postmenstrual age (PMA), head circumference (HC), retinopathy of prematurity (ROP), intraventricular hemorrhage (IVH), necrotizing enterocolitis (NEC), bronchopulmonary dysplasia (BPD).

**Table 3 nutrients-15-04158-t003:** Parenteral and enteral nutrition.

Variables	AGA Group (*n* = 249)	IUGR Group (*n* = 40)	SGA Group (*n* = 11)	IUGR vs. AGA *p*-Values	SGA vs. AGA *p*-Values
Parenteral and Enteral Nutrition *					
Days on PN	18 (12, 21)	17 (11, 21)	24 (16, 33)	0.68	0.07
Total Energy (kcal/kg/d)	108 (104, 116)	107 (104, 113)	114 (110, 120)	0.31	0.14
Total Carbohydrates (mg/kg/min)	7.9 (6.8, 8.6)	7.4 (6.2, 8.3)	6.4 (5.9, 7.9)	0.73	0.29
Total Protein (g/kg/d)	3.7 (3.5, 3.8)	3.6 (3.4, 3.6)	3.3 (3.1, 3.9)	0.90	0.14
Total Fat (g/kg/d)	3.9 (3.6, 4.5)	3.5 (3.0, 4.2)	3.4 (2.8, 4.1)	0.43	0.51
Enteral nutrition, % (*n*)					
Human Milk + HMF	28 (70/249)	30 (12/40)	27 (3/11)	0.80	0.95
Human Milk + BMF	33 (82/249)	30 (12/40)	45 (5/11)	0.71	0.38
Formula	5 (12/249)	7 (3/40)	0 (0/11)	0.47	0.45
Mixed	34 (85/249)	33 (13/40)	27 (3/11)	0.83	0.63
Exclusively mother’s own milk at discharge	61 (152/249)	60 (24/40)	73 (8/11)	0.68	0.45

* Values are median (interquartile range). Human milk-based fortifier (HMF), bovine milk-based fortifier (BMF).

**Table 4 nutrients-15-04158-t004:** Unadjusted anthropometric and body composition parameters at term-equivalent age.

	AGA Group (*n* = 249)	IUGR Group (*n* = 40)	SGA Group (*n* = 11)
PMA at measurement, week	42.1 (40.0, 46.3)	43.0 (40.5, 46.9)	42.1 (39.4, 52.6)
Body composition parameters			
FFM, percentage	77.8 (73.9, 83.6)	81.9 (73.3, 85.4)	78.2 (76.6, 81.2)
FM, percentage	22.2 (16.5, 26.1)	18.2 (14.6, 26.7)	21.8 (18.8, 23.4)
FFM, gram	2896 (2403, 3412)	2428 (2147, 2859)	2510 (2121, 2920)
FM, gram	785 (512, 1174)	662 (382, 893)	620 (530, 799)
Anthropometric parameters			
Weight, gram	3681 (2915, 4586)	3090 (2529, 3752)	3130 (2651, 3719)
Length, cm	46.0 (44.0, 48.5)	44.0 (41.0, 46.0)	44.0 (40.0, 46.0)
Head circumference, cm	33.0 (31.5, 34.0)	32.0 (30.8, 33.8)	31.5 (30.7, 33.8)

Data are median and Interquartile Range (IQR). Postmenstrual age (PMA).

**Table 5 nutrients-15-04158-t005:** Adjusted weight and body composition values for AGA, IUGR, and SGA groups.

	Adjusted Mean			Adjusted Mean Difference	
	AGA Group	IUGR Group	SGA Group	IUGR Group	SGA Group
**Total (*n*)**	**249**	**40**	**11**		
Weight at scan, gram ^1^	3797 (3745, 3849)	3667 (3528, 3806)	3736 (3480, 3992)	−130 (−21, 281)	−46 (−215, 307)
				*p* = 0.09	*p* = 0.73
FFM, Z-Score	−1.1 (−1.2, −1.0)	−1.5 (−1.8, −1.2)	−1.0 (−1.6, −0.5)	−0.4 (−0.8, −1.0)	−0,1 (−0.5, 0.7)
				*p* = 0.017	*p* = 0.79
FM, Z-Score	0.8 (0.7, 0.9)	0.7 (0.3, 1.0)	1.2 (0.5, 1.9)	0.1 (−0.5, 0.3)	0.4 (−0.3, 1.1)
				*p* = 0.48	*p* = 0.24
FFM, gram	2977 (2919, 3035)	2654 (2502, 2807)	2823 (2690, 2956)	−323 (−488, −158)	−154 (−229, 79)
				*p* < 0.001	*p* = 0.09
FM, gram	843 (817, 868)	847 (774, 921)	958 (825, 1092)	4 (8, 53)	116 (−21, 252)
				*p* = 0.99	*p* = 0.09

^1^ Mean (95% CI) adjusted for sex, gestational age at birth, and postmenstrual age, weight, and length at measurement.

## Data Availability

The data presented in this study are available on request from the corresponding author. The data are not publicly available due to ongoing research.

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
