# Peer review of "Effects of Intrauterine Growth Restriction (IUGR) on Growth and Body Composition Compared to Constitutionally Small Infants"

_nutrients, 2023, doi:10.3390/nu15194158_

Round 1

Reviewer 1 Report

Calek et al conducted a retrospective cohort study to evaluate growth and body composition at term-equivalent age in IUGR, constitutionally small infants (SGA), and appropriate for gestational age (AGA) preterm infants. The primary results include a slower growth trajectory and a reduction in fat free mass (FFM) in IUGR infants compared to both AGA and SGA infants. There were no differences in fat mass (FM) among groups. Strengths of the study are that the definition and inclusion criteria for IUGR is stringent and well defined; all infants had Peapod assessment for body composition; and addresses an important question about how the growth of preterm infants varies regarding growth trajectory and body composition among IUGR, SGA and AGA groups. The paper is generally well written, and several key issues relating to growth of the preterm infant based on intrauterine factors are addressed. Weaknesses are as follows.

1.      There was no primary outcome defined, nor was there a power calculation reported. The lack of power calculation is particularly relevant because several statements are made that AGA and SGA infants did not differ in body composition or growth. However, with only an n=11 of SGA infants with wide variability in gestational age, birthweight, and premature birth-related outcomes, these conclusions likely suffer from Type II error.

2.     It appears that infants were included in the study if PEAPOD body composition was performed. It is stated that it is standard practice to perform PEAPOD on all preterm infants at term equivalent gestational age. Were there infants during that time period who did not have PEAPOD, such that results may have been biased based on which infants could have PEAPOD performed compared to those that did not (i.e. oxygen requirement, parental refusal, etc)? A flow chart would be helpful.

3.     A fundamental problem is that the gestational ages among the three groups varied widely and significantly. Surprisingly, the IUGR infants were born at a more mature gestational age compared to the other groups, when it would be expected that IUGR infants would be delivered more prematurely given distress in utero. Given that this is a retrospective study design, was it considered to perform a retrospective case control study so that infants could be matched based on gestational age? In the current study design, gestational age is a significant confounding factor.

4.     A preferred statistical analysis plan when comparing baseline characteristics, growth velocities, short term outcomes, etc would be to perform an ANOVA with posttest comparisons, as opposed to performing individual t-test or Mann-Whitney tests between AGA and IUGR, and AGA and SGA.

5.     It would be helpful to present the mean caloric intake per day and other nutritional intake amounts, as this may explain the reduction in growth velocity in the IUGR group. This is especially relevant given the significant discussion in lines 262-276 and elsewhere about how nutritional delivery is essential to avoid extrauterine growth restriction. If there were no differences in nutritional delivery among groups, it would support that programming of growth in utero may be the dominant factor in limiting postnatal growth.

6.     It is not clear why a linear regression model was used for FM and FFM Z-score comparisons, and not for FM and FFM comparisons.

7.     Regarding the linear regression model that birth weight Z-score was adjusted for and not total weight at scan. Birthweight z-score would be in the causal pathway when comparing AGA, SGA and IUGR groups. Total weight at scan would be more appropriate to adjust for, as body weight may influence body composition. The other covariates that were adjusted for are appropriate.

8.     Lines 301-302 are incorrect. According to Table 1, birthweight was not lower in the IUGR group compared to the AGA group. In fact, it was significantly higher (p=0.05).

9.     Though the paper has several limitations as noted above, none are acknowledged or discussed (with the exception of small sample size).

10.  The manuscript requires careful editing, especially related to verb tense.

As noted, careful editing for verb tense is necessary.

Author Response

Dear Reviewer,

we would like to thank you for the very constructive criticism that helped to improve our manuscript.

Changes are marked in the point-by-point reply while the manuscript is a version with track changes.

Response to Reviewer 1 Comments

Authors’ comments: We would like to thank the reviewer for the very constructive criticism that helped to improve our manuscript.

Changes are marked in the point-by-point reply while the manuscript is a version with track changes.

Reviewer comments: Calek et al conducted a retrospective cohort study to evaluate growth and body composition at term-equivalent age in IUGR, constitutionally small infants (SGA), and appropriate for gestational age (AGA) preterm infants. The primary results include a slower growth trajectory and a reduction in fat free mass (FFM) in IUGR infants compared to both AGA and SGA infants. There were no differences in fat mass (FM) among groups. Strengths of the study are that the definition and inclusion criteria for IUGR is stringent and well defined; all infants had Peapod assessment for body composition; and addresses an important question about how the growth of preterm infants varies regarding growth trajectory and body composition among IUGR, SGA and AGA groups. The paper is generally well written, and several key issues relating to growth of the preterm infant based on intrauterine factors are addressed. Weaknesses are as follows.

Point 1: There was no primary outcome defined, nor was there a power calculation reported. The lack of power calculation is particularly relevant because several statements are made that AGA and SGA infants did not differ in body composition or growth. However, with only an n=11 of SGA infants with wide variability in gestational age, birth weight, and premature birth-related outcomes, these conclusions likely suffer from Type II error.

Response 1: We thank the reviewer for the constructive comment and revised the manuscript. We have added new information in the method and results section and highlighted that this is an exploratory study. Consequently, a study with larger sample size is necessary to draw final conclusions. However, this is the first study that analysed growth and body composition in IUGR- and constitutionally small infants compared to AGA infants. These results might have a relevant impact on future nutritional care and strategies in constitutionally small infants.

Method section:

Line 87-89:

“This exploratory retrospective cohort study was performed at the Department of Pediatrics and Adolescent Medicine, Division for Neonatology, Paediatric Intensive Care and Neuropediatric at the Medical University of Vienna, Austria.”

Discussion section:

Line 446-448:

“Furthermore, these data are generated from an exploratory study and the analysis requires cautious interpretation. Further studies with a larger sample size and power calculation are necessary to draw final conclusions.” 

Point 2: It appears that infants were included in the study if PEAPOD body composition was performed. It is stated that it is standard practice to perform PEAPOD on all preterm infants at term equivalent gestational age. Were there infants during that time period who did not have PEAPOD, such that results may have been biased based on which infants could have PEAPOD performed compared to those that did not (i.e. oxygen requirement, parental refusal, etc)? A flow chart would be helpful.

Response 2: Agreed. We thank the reviewer for the constructive comment and added new information in the results section.

After discharge, preterm infants are followed up through our outpatient clinic. As part of their regular check-ups, we consistently conduct body composition measurements. Nasal cannula, percutaneous endoscopic gastrostomy tubes, feeding tubes, stoma bags, ventriculoperitoneal shunts or other essential, non-removable medical devices can also undergo pre-calibration to be factored into the body composition assessment. While only a small number of infants are discharged with oxygen, most of them handle short breaks of oxygen-supported breathing quite well, so the body composition measurement, which can be done quickly, is not an obstacle. However, in only 5 cases body composition was not performed and a bias seems therefore unlikely. We added new information in the results section.

Results section:

Line 189-191:

“Body composition measurement was not performed in four infants due to following conditions: continuous oxygen requirement: n=1, IUGR-group; hemodynamic instability: n=1, AGA-group; and loss of follow up: n=2, AGA-group (Flowchart 1).”

Line 203

Flowchart 1. Overview of the study groups

Point 3: A fundamental problem is that the gestational ages among the three groups varied widely and significantly. Surprisingly, the IUGR infants were born at a more mature gestational age compared to the other groups, when it would be expected that IUGR infants would be delivered more prematurely given distress in utero. Given that this is a retrospective study design, was it considered to perform a retrospective case control study so that infants could be matched based on gestational age? In the current study design, gestational age is a significant confounding factor.

Response 3: We thank the reviewer for the constructive comments.

Gestational age was significantly higher in the IUGR group than in the AGA group. We considered a matching process, but the sample size is relatively small and gestational age at birth varied widely. Therefore, it was not possible and problematic (loss of cases) to match the groups. In the multivariable regression analysis, we incorporated gestational age at birth as a covariate.

Discussion section:

Line 440-448:

One limitation of the study is the relatively small sample size and the retrospective nature of this study. We considered a matching process, but the sample size in this study is relatively small, because being constitutionally small is an exceedingly rare condition and gestational age at birth varied widely. Therefore, we incorporated gestational age at birth as a covariate in the multivariable regression analysis. However, designing and conducting a clinical trial with an adequate number of constitutionally small infants would be very challenging. Furthermore, these data are generated from an exploratory study and the analysis requires cautious interpretation. Further studies with a larger sample size and power calculation are necessary to draw final conclusions.

Point 4: A preferred statistical analysis plan when comparing baseline characteristics, growth velocities, short term outcomes, etc would be to perform an ANOVA with posttest comparisons, as opposed to performing individual t-test or Mann-Whitney tests between AGA and IUGR, and AGA and SGA.

Response 4:

We agree that ANOVA is recommended in general for any comparison of several groups. If the result of the test is 'not significant' then it is commonly concluded that the groups (IUGR, AGA and SGA in our case) do not differ, even though the appropriate conclusion would be that 'we have no evidence of any differences among the groups.

Point 5: It would be helpful to present the mean caloric intake per day and other nutritional intake amounts, as this may explain the reduction in growth velocity in the IUGR group. This is especially relevant given the significant discussion in lines 262-276 and elsewhere about how nutritional delivery is essential to avoid extrauterine growth restriction. If there were no differences in nutritional delivery among groups, it would support that programming of growth in utero may be the dominant factor in limiting postnatal growth.

Response 5: Agreed. We thank the reviewer for the constructive comments and added new information in the method, results, and discussion section. The nutritional management was not significantly different between the study groups. 

Method section:

Line 167-170: Enteral nutrition (mother´s own milk, fortification, formula, and mixed nutrition) at discharge and parenteral nutritional supply (kilocalories kcal/kg/d, carbohydrates mg/kg/min, proteins g/kg/d, and fat g/kg/d) are reported.

Line 180-183:

Mann-Whitney-U and Pearson-Chi-Square tests were used to compare baseline characteristics, growth velocity (grams/kg/day from birth until discharge), nutritional management (parenteral and enteral nutrition), and co-morbidities (IVH, NEC, ROP, BPD, culture-proven sepsis).

Result section

Line 278-281:

Enteral and parenteral nutritional supply (kcal, carbohydrates, protein, and fat) were not significantly different between the groups (Table 3.). Enteral nutrition at discharge was not significantly between the groups (Table 3.).

Line 284

Table 3. Parenteral and enteral nutrition

Variables

AGA group

(n= 249)

IUGR group

(n= 40)

SGA group

(n= 11)

IUGR vs. AGA p-values

SGA vs. AGA

p-values

Parenteral and Enteral Nutrition*

Days on PN

18 (12, 21)

17 (11, 21)

24 (16, 33)

0.68

0.07

Total Energy (kcal/kg/d)

108 (104, 116)

107 (104, 113)

114 (110, 120)

0.31

0.14

Total Carbohydrate (mg/kg/min)

7.9 (6.8, 8.6)

7.4 (6.2, 8.3)

6.4 (5.9, 7.9)

0.73

0.29

Total Protein (g/kg/d)

3.7 (3.5, 3.8)

3.6 (3.4, 3.6)

3.3 (3.1, 3.9)

0.90

0.14

Total Fett (g/kg/d)

3.9 (3.6, 4.5)

3.5 (3.0, 4.2)

3.4 (2.8, 4.1)

0.43

0.51

Enteral nutrition, % (n)

Human Milk + HMF

28 (70/249)

30 (12/40)

27 (3/11)

0.68

0.87

Human Milk + BMF

33 (82/249)

30 (12/40)

45 (5/11)

0.84

0.22

Formula

5 (12/249)

7 (3/40)

0

0.44

-

Mixed

 34 (85/249)

33 (13/40)

27 (3/11)

0.98

0.87

Exclusively mother´s own milk at discharge

61 (152/249)

60 (24/40)

73 (8/11)

0.68

0.45

* Values are median (interquartile range), human milk-based fortifier HMF), bovine milk-based fortifier (BMF)

Discussion section:

Line 390-399:

The nutritional management, nutrient supply, and days on parenteral nutrition during hospital stay were not significantly different between the three study groups. Furthermore, the rates of exclusively mother´s own milk at discharge were not significantly different between the groups (between 70-80%). These data support the hypothesis that the nutritional supply was inadequate to support normal growth in infants with IUGR. Consequentially, individualized fortification may improve growth and body composition in infants with IUGR. However, further research is needed to investigate if the nutritional supply during pregnancy and consequently growth in utero may be the dominant factor in limiting postnatal growth and body composition.

Point 6: It is not clear why a linear regression model was used for FM and FFM Z-score comparisons, and not for FM and FFM comparisons.

Response 6: Agreed. We also analyzed FM gram and FFM gram.

We added new information in the method, results (Table 5), and discussion section.

FFM- and FM grams were also not significantly different between the SGA- and AGA groups. FFM gram was significantly lower in the IUGR- compared to the AGA groups.

Method section:

Line 175:

A multivariable regression analysis was used to investigate the association between body composition (FFM gram and FM gram and FFM- and FM-Z-Scores) and body weight […].

Results section:

Line 297-299:

FFM-Z-score and FFM gram were significantly lower in infants with IUGR in comparison to AGA infants (p = 0.017; p <0.001, respectively), while there were no significant difference in FFM-Z-Score and FFM gram between SGA and AGA infants (p = 0.78, p = 0.09, respectively) (Table 5). FM-Z-Scores and FM gram were not significantly different between the IUGR- and AGA groups and SGA- and AGA groups (Table 5).

Table 5. Adjusted weight and body composition values for AGA, IUGR, and SGA groups.

Adjusted mean

    Adjusted mean difference

AGA group

IUGR group

SGA group

IUGR group

 SGA group

Total (n)

249

40

11

Weight at scan, gram1

3797 (3745, 3849)

3667 (3528, 3806)

3736 (3480, 3992)

-130 (-21, 281)

p = 0.09

 -46 (-215, 307)

   p = 0.73

FFM, Z-Score

-1.1 (-1.2, -1.0)

-1.5 (-1.8, -1.2)

-1.0 (-1.6, -0.5)

   -0.4 (-0.8, -1.0)

   p = 0.017

    -0,1 (-0.5, 0.7)

 p = 0.79

 FM, Z-Score

0.8 (0.7, 0.9)

0.7 (0.3, 1.0)

1.2 (0.5, 1.9)

   0.1 (-0.5, 0.3)

  p = 0.48

 0.4 (-0.3, 1.1)

   p = 0.24

 FFM, gram

2977 (2919, 3035)

2654 (2502, 2807)

 2823 (2690, 2956)

-323 (-488, -158)

-154 (-229, 79)

 p <0.001

p = 0.09

 FM, gram

 843 (817, 868)

847 (774, 921)

958 (825, 1092)

4 (8, 53)

116 (-21, 252)

p =0.99

p =0.09

1Mean (95% CI) adjusted for sex, gestational age at birth, and postmenstrual age, weight, and length at measurement.

Discussion section:

Line 313-316:

FFM-Z-Score and FFM gram were significantly lower in infants with IUGR in comparison to AGA infants, while there were no significant differences in FFM-Z-Scores and FFM gram between SGA- and AGA infants. FM-Z-Scores and FM gram were not significantly different in IUGR- and SGA infants compared to AGA infants.

Point 7: Regarding the linear regression model that birth weight Z-score was adjusted for and not total weight at scan. Birth weight z-score would be in the causal pathway when comparing AGA, SGA and IUGR groups. Total weight at scan would be more appropriate to adjust for, as body weight may influence body composition. The other covariates that were adjusted for are appropriate.

Response 7: Agreed. We have added “weight at measurement” in the linear regression model and removed birth weight z-score as requested. We recalculated and found no significant changes.

Method section:

Line 175-180:

A multivariable regression analysis was used to investigate the association between body composition (FFM gram and FM gram and FFM- and FM-Z-Scores) and body weight at term-equivalent age and the study groups (AGA, IUGR, and SGA) while adjusting for the covariates, including sex [44], gestational age at birth [45], and age, weight, and length at measurement to adjust for body size [45].

Line 307

Table 5.

1Mean (95% CI) adjusted for sex, gestational age at birth, and postmenstrual age, weight, and length at measurement.

Point 8: Lines 301-302 are incorrect. According to Table 1, birth weight was not lower in the IUGR group compared to the AGA group. In fact, it was significantly higher (p=0.05).

Response 8: We thank the referee for careful reading. We must apology for the mistake. We have corrected the sentence as follows:

Line 379:

“In our study, weight at birth was significantly lower in SGA infants in comparison to AGA infants.”

Point 9: Though the paper has several limitations as noted above, none are acknowledged or discussed (with the exception of small sample size).

Response 9: Agreed. We have added and discussed further limitations in the discussion section.

Line 441-448:

One limitation of the study is the relatively small sample size and the retrospective nature of this study. We considered a matching process, but the sample size in this study is relatively small, because being constitutionally small is an exceedingly rare condition and gestational age at birth varied widely. Therefore, we incorporated gestational age at birth as a covariate in the multivariable regression analysis. However, designing and conducting a clinical trial with an adequate number of constitutionally small infants would be very challenging. Furthermore, these data are generated from an exploratory study and the analysis requires cautious interpretation. Further studies with a larger sample size and power calculation are necessary to draw final conclusions.

Point 10: The manuscript requires careful editing, especially related to verb tense.

Response 10: Agreed. We thank the reviewer for the constructive point.

A native English speaker has reviewed the manuscript and revised the manuscript.

Reviewer 2 Report

The paper entitled "Effects of intrauterine growth restriction (IUGR) compared to constitutionally small infants on growth and body composition" has great potential to be published in Nutrients. However, it is poorly written. The structure of some sentences is unusual and it is difficult to follow the paper. Authors need to edit the text and rewrite some parts with the help of an English-speaking person.

Major comments:

1) Authors need to add a table describing the criteria of experimental groups in the "Method section".

2)In the Introduction section, the Authors need to discuss the mechanism of growth restriction: PMID: 37186255; PMID: 33259808, etc

3)  Does authors find any differences between males and females? Authors must include this information for each experimental group.

The structure of some sentences is unusual and it is difficult to follow the paper. Authors need to edit the text and rewrite some parts with the help of an English-speaking person. I recommend rewriting the Abstract and Conclusion

Author Response

Dear Reviewer,

we would like to thank you for the very constructive criticism that helped to improve our manuscript.

Changes are marked in the point-by-point reply while the manuscript is a version with track changes.

Response to Reviewer 2 Comments

Authors’ comments: We would like to thank the reviewer for the very constructive criticism that helped to improve our manuscript.

Changes are marked in the point-by-point reply while the manuscript is a version with track changes.

Reviewer comments: The paper entitled "Effects of intrauterine growth restriction (IUGR) compared to constitutionally small infants on growth and body composition" has great potential to be published in Nutrients. However, it is poorly written. The structure of some sentences is unusual and it is difficult to follow the paper. Authors need to edit the text and rewrite some parts with the help of an English-speaking person.

Point 1: Authors need to add a table describing the criteria of experimental groups in the "Method section".

Response 1: Agreed. We thank the reviewer for the constructive comments, and we revised the manuscript as requested. The criteria for the study groups are stated in the method section and we added a new flowchart in the results section for better understanding.

Method section:

Group assignment (AGA, IUGR, and SGA groups) was based on weight percentile and signs of placental insufficiency determined by prenatal ultrasound [17, 33]. In singleton pregnancies, IUGR was diagnosed according to the consensus definition by Gordjin et al. [17], which is defined as follows: 1) estimated fetal weight (EFW) <10 percentile and 2) uterine artery pulsatility index (Uta-PI) and/or umbilical artery PI (UA-PI) >95th percentile and/or middle cerebral artery (MCA-PI) <5th percentile. Constitutionally small (SGA group) was defined as infants with a birth weight <10th percentile and without any signs of placental insufficiency determined by prenatal ultrasound. AGA was defined as birth weight between the 10th and the 90th percentile without Doppler ultrasound abnormalities [17].

Line 203

Flowchart 1. Overview of the study groups

Point 2: In the Introduction section, the Authors need to discuss the mechanism of growth restriction: PMID: 37186255; PMID: 33259808, etc

Response 2: We thank the reviewer for the constructive comments and added new information and references [19-22] in the introduction.

Introduction section:

Line 53-61:

IUGR is defined as a fetus that has failed to reach its growth potential, which can be due to several conditions, but is primarily caused by placental insufficiency and is characterized by impaired oxygen and nutritional supply [17, 18]. The underlying cellular and molecular mechanism of fetal growth restriction is still not fully investigated so far [17, 19]. Placental-mediated fetal growth restriction arises mainly from maldevelopment of the placental vascular system [20]. Several studies showed that impaired fetoplacental angiogenesis is associated with fetal growth restriction [21, 22]. Furthermore, recent research investigated novel integrin-extracellular matrix interactions that regulate placental angiogenesis in severe fetal growth restriction [19]. However, further research is needed to investigate the underlying cause of fetal growth restriction as well as prevention and treatment strategies.

References:

  1. Diane L Gumina, Emily J Su. Mechanistic insights into the development of severe fetal growth restriction. lin Sci (Lond).2023 Apr 26;137(8):679-695.doi: 10.1042/CS20220284.

  1. Graham J Burton, Eric Jauniaux. Pathophysiology of placental-derived fetal growth restrictionm J Obstet Gynecol.2018 Feb;218(2S):S745-S761.doi: 10.1016/j.ajog.2017.11.577.
  2. Emily J Su, Hong Xin, Ping Yin, Matthew Dyson, John Coon, Kathryn N Farrow, Karn K Mestan, Linda M Ernst. Impaired fetoplacental angiogenesis in growth-restricted fetuses with abnormal umbilical artery doppler velocimetry is mediated by aryl hydrocarbon receptor nuclear translocator (ARNT). J Clin Endocrinol Metab. 2015 Jan;100(1):E30-40.doi: 10.1210/jc.2014-2385.
  3. Shuhan Ji, Hong Xin, Emily J Su. Overexpression of the aryl hydrocarbon receptor nuclear translocator partially rescues fetoplacental angiogenesis in severe fetal growth restriction. Sci (Lond).2019 Jun 20;133(12):1353-1365.doi: 10.1042/CS20190381. Print 2019 Jun 28.

Point 3:  Does authors find any differences between males and females? Authors must include this information for each experimental group.

Response 3:

We thank the reviewer for the constructive comments, and we revised the manuscript as requested.

Gale et al. [48] investigated sexual dimorphism in relation to adipose tissue in early infancy. Therefore, we adjust for sex in the multiple regression analysis. We found that sex was not significantly influencing growth and body composition. We hypothesized that the sample size was too small (SGA-group: female n=4 and IUGR-group: male n=14) to investigate differences.

Results section:

Line 301-306:

Sex did not significantly influence weight and body composition. Weight: IUGR versus AGA; p = 0.12 and SGA versus AGA; p = 0.23. FFM gram: IUGR versus AGA; p = 0.12 and SGA versus AGA; p = 0.10. FFM-Z-Score: IUGR versus AGA; p = 0.06 and SGA versus AGA; p = 0.07. FM-gram: IUGR versus AGA; p = 0.24 and SGA versus AGA; p = 0.21. FM-Z-Score: IUGR versus AGA; p = 0.13 and SGA versus AGA; p = 0.14.

Reference:

  1. Gale, C., et al., Sexual dimorphism in relation to adipose tissue and intrahepatocellular lipid deposition in early infancy. Int J Obes (Lond), 2015. 39(4): p. 629-32.

Comments on the Quality of English Language

Point 4: The structure of some sentences is unusual and it is difficult to follow the paper. Authors need to edit the text and rewrite some parts with the help of an English-speaking person. I recommend rewriting the Abstract and Conclusion

Response 4: Agreed. We thank the reviewer for the constructive point.

A native English speaker has reviewed the manuscript and revised the manuscript. We also rewrote the abstract and conclusion sections.

Reviewer 3 Report

This is an important research as it demonstrated the differences of being IUGR and constitutionally small at birth. Despite having a small number of IUGR and SGA newborns in this sample, the differences were still significant,  mainly regarding growth velocity and free fat mass at term age in those newborns with IUGR - which means that these are robust differences.

Author Response

Dear Reviewer,

we would like to thank you for the very constructive criticism that helped to improve our manuscript.

Round 2

Reviewer 2 Report

I recommend accepting this manuscript in its present form.

Author Response

We thank the reviewer for the comment that helped to improve our manuscript.